


# Global Evaluation of the Ecosystem Demography Model (ED
# v3.0)
Lei Ma[1], George Hurtt[1], Lesley Ott[2], Ritvik Sahajpal[1], Justin Fisk[3], Rachel Lamb[1], Hao Tang[1,7],
Steve Flanagan[4], Louise Chini[1], Abhishek Chatterjee[2,5], Joseph Sullivan[6]
[1] Department of Geographical Sciences, University of Maryland, College Park, MD 20770, USA
[2] Global Modeling and Assimilation Office, NASA Goddard Space Flight Center, Greenbelt, MD 20771, USA
[3] Regrow Agriculture Inc., Durham, NH 03824, USA
[4] Wildland Fire Science, Tall Timbers Research Station and Land Conservancy, Tallahassee, FL 32312, USA
[5] Universities Space Research Association, Columbia, MD 21046, USA
[6] Department of Plant Science & Landscape Architecture, University of Maryland, College Park, MD 20770, USA
[7] Department of Geography, National University of Singapore, 117570, Singapore
*Correspondence to*: Lei Ma (lma6@umd.edu)
**Abstract.**
Terrestrial ecosystems play a critical role in the global carbon cycle but have highly uncertain future dynamics.
Ecosystem modelling that includes the scaling-up of underlying mechanistic ecological processes has the potential
to improve the accuracy of future projections, while retaining key process-level detail. Over the past two decades,
multiple modelling advances have been made to meet this challenge, including the Ecosystem Demography (ED)
model and its derivatives including ED2 and FATES. Here, we present the global evaluation of the Ecosystem
Demography model (ED v3.0), which likes its predecessors features the formal scaling of physiological processes of
individual-based vegetation dynamics to ecosystem scales, together with integrated submodules of soil
biogeochemistry and soil hydrology, while retaining explicit tracking of vegetation 3-D structure. This new version
builds on previous versions and provides the first global calibration and evaluation, global tracking of the effects of
climate and land-use change on vegetation 3-D structure, new spin-up process and input datasets, as well as
numerous other advances. Model evaluation was performed with respect to a set of important benchmarking
datasets, and model estimates were within observational constraints for multiple key variables including: (i) global
patterns of dominant plant functional types (broadleaf vs evergreen); (ii) spatial distribution, seasonal cycle, and
interannual trends of global Gross Primary Production (GPP); (iii) global interannual variability of Net Biome
Production (NBP); and (iv) global patterns of vertical structure including leaf area and canopy height. With this
global model version, it is now possible to simulate vegetation dynamics from local to global scales and from



seconds to centuries, with a consistent mechanistic modelling framework amendable to data from multiple
traditional and new remote sensing sources, including lidar.
**1 Introduction**
Terrestrial ecosystems and the associated carbon cycle are of critical importance in providing ecosystem services
and regulating global climate. Plants store approximately 450-650 Pg C as biomass globally. They remove
approximately 120 Pg C from the atmosphere each year through photosynthesis, and release a similar magnitude of
carbon to the atmosphere through respiration (Beer et al., 2010; Ciais et al., 2014a). Human activities over past
centuries have significantly impacted terrestrial ecosystems through biophysical and biogeochemical mechanisms
(Cramer et al., 2001; Walther et al., 2002; Brovkin et al., 2004; Pielke Sr. et al., 2011). Quantification, attribution
and future projections of the terrestrial carbon sink requires in-depth understanding of underlying ecological
processes and their sophisticated responses and feedbacks to climate change, elevated $CO_2$, and land use and land
cover change (LULCC) across multiple biomes and spatial and temporal scales (Canadell et al., 2007; Erb et al.,
2013; Keenan and Williams, 2018). This demand for information has driven the emergence and development of
dynamic global ecosystem models (DGVMs), which simplify the structure and functioning of global vegetation into
several plant functional types and simulate vegetation distribution and associated biogeochemical and hydrological
cycles with ecophysiological principles (Prentice et al., 2007; Prentice and Cowling, 2013). The first generation of
DGVMs have been used successfully to address a variety of carbon cycle related questions and integrated into Earth
System Models (ESMs) (Cramer et al., 2001; Sitch et al., 2008). Subsequent developments have improved the
representation of vegetation demographic processes within ESMs, including the Ecosystem Demography model
(ED) (Hurtt et al., 1998; Moorcroft et al., 2001), ED2 (Medvigy 2006; Medvigy et al., 2009; Longo et al., 2019a),
CLM(ED) (Fisher et al., 2015; Lawrence et al., 2019; Massoud et al., 2019),  SEIB-DFVM (Spatially-Explicit
Individual-based Dynamic Global Vegetation Model) (Sato et al ., 2007), LPJ-GUESS (Lund-Postdam-Jena General
Ecosystem Simulator) (Smith et al., 2001, 2014) and GFDL-LM3-PPA (Geophysical Fluid Dynamics Laboratory
Land Model 3 with the Perfect Plasticity Approximation) (Weng et al., 2015), as summarized in Fisher et al., 2018.

In addition to model development, model evaluation is an important process to assess model uncertainties and also
identify processes that need particular improvements (Anav et al., 2013; Luo et al., 2012; Eyring et al., 2019).
Considerable effort has been spent on standardizing evaluation practices and developing a comprehensive
benchmarking system (Abramowitz et al., 2012; Collier et al., 2018; Eyring et al., 2016; Kelly et al., 2013;
Randerson et al., 2009). For example, a benchmarking system from the International Land Model Benchmarking
(ILAMB) project has been increasingly used to evaluate ecosystem and climate models (Collier et al., 2018;
Ghimire et al., 2016; Luo et al., 2012). In parallel, new observations are providing new opportunities to initialize and
test models. Of particular relevance for ecosystem models is the advent of spaceborne lidar missions (i.e., GEDI and
ICESat-2) (Dubayah et al., 2020a; Markus et al., 2017), which provide unprecedented global observations of forest
structure, including vertical distribution of leaf foliage. Building on this past work, and utilizing new observations,
an updated and systematic evaluation of model performance on multiple variables is now possible.




Here, we present the global evaluation of Ecosystem Demography v3.0. The ED model was developed two decades
ago using a formal scaling approach (Size- and Age-Structured approximation, SAS) to efficiently approximate the
expected dynamics of individual based forest dynamics (Hurtt et al., 1998; Moorcroft et al., 2001). Since its
emergence, the ED model has been continuously developed and applied at various regions and spatial scales, with
land-use changes, and lidar observations (Hurtt et al., 2002, 2004). In the original paper, the model was implemented
at site scale and primarily evaluated for aboveground biomass accumulation during succession using
chronosequence field data, and at the regional scale using 1-degree resolution data on potential biomass, soil carbon
and net primary productivity (NPP) (Moorcroft et al., 2001). Most recently, ED was implemented at high spatial
resolution (90 m) over a regional domain of the Northeastern United States and evaluated for aboveground biomass
using wall-to-wall lidar-based estimates of contemporary biomass at that spatial resolution (Hurtt et al., 2019a; Ma
et al., 2021). The evaluation included >30 million grid cell pairs, and >$10^3$ forest inventory plots. This progression
spans a range of model capabilities, spatial resolutions, and evaluation data ranging from coarse resolution potential
vegetation, to high spatial resolution contemporary conditions at regional scales. However, development and
evaluation of ED at global scale for contemporary conditions has not yet been accomplished. In this study, ED v3.0
is evaluated at global scales for the first time. Multiple key variables are considered in the evaluation, including
benchmark datasets on vegetation distribution, vegetation structure, and carbon and water fluxes.
**2 Methods**
ED v3.0 is built upon a series of previous model developments (Moorcroft et al., 2001; Hurtt et al., 2002; Albani et
al., 2006; Fisk, 2015; Flanagan et al., 2019). To extend ED's capabilities globally, several additional modifications
were introduced to capture global vegetation distribution across biomes and related carbon stocks and fluxes. Below,
a summary of the ED approach and recent modifications is provided. The full descriptions of each submodule can be
found in the Supplement along with tables of parameter values. To conduct the model evaluation, a model
experimental protocol including equilibrium and transient simulations was developed and relevant forcing data were
identified from global existing datasets. Model simulations were then compared to benchmarking datasets.
**2.1 Model**
The ED model is an individual based prognostic ecosystem model (Moorcroft et al., 2001). By integrating
submodules of growth, mortality, hydrology, carbon cycle, and soil biogeochemistry, ED can track plant dynamics
including growth, mortality, and reproduction. Along with plant dynamics, ED can track the carbon cycle, including
carbon uptake by leaf photosynthesis, carbon allocation to biomass growth in leaves, roots and stems, carbon
redistribution from plants to soil based on plant tissue turnover from dead plants due to mortality and disturbance,
carbon decomposition in various pools (metabolic litter pool, structural litter pool, soil slow pool, soil passive pool,
wood product pool, harvested crop pool, etc) as well as carbon combustion from fire (Fig. 1 and Fig. 2). Over the
last two decades, ED has been continuously developed and combined with lidar and land-use change data to predict





ecosystem dynamics and associated water and carbon fluxes across spatial scales (e.g., site to regional and
continental) and temporal scales (e.g., short-term seasonal to long-term decadal and century) (Hurtt et al., 2002,
2004, 2010, 2016, Fisk et al., 2013, Flanagan et al., 2019). ED distinguishes itself from most other ecosystem
models by explicitly tracking vegetation structure and scaling fine-scale physiological processes to large scale
ecosystem dynamics (Hurtt et al., 1998, Moorcroft et al., 2001, Fisher et al., 2018). In ED, vegetation structure (e.g.,
height and diameter at breast height), and physiological processes (e.g., leaf photosynthesis and phenology) are
modelled at the individual scale, where individual plants compete mechanically for light, water, and nutrients.
During implementation, this horizontal heterogeneity is tracked through cohort and patch demography. Explicitly
modelling vegetation height facilitates a potential connection to lidar data. The most advanced version of ED was
used in this study and it has been recently calibrated and evaluated globally by various benchmarking datasets such
as gross primary productivity (GPP), leaf area index (LAI), aboveground biomass (AGB), and net biome
productivity (NBP) (Ma et al., 2021).
**2.1.1 Additional modifications**
Major modifications in ED v3.0 focus on four areas: plant functional type representation, leaf level physiology,
hydrology, and wood products. These areas have been found to be particularly important to improve model
performance globally.

Plant functional types describe the characteristics of vegetation in different representative groups for modelling. In
previous ED versions, various PFT combinations were implemented to represent vegetation in the respective regions
where the model was implemented. In the original implementation of ED for Central and South America, four PFTs
were represented (i.e., Early-successional broadleaf, Middle-successional broadleaf, Late-successional broadleaf and
C4 grasses (Moorcroft et a., 2001). In a subsequent implementation over North America, two additional PFTs (i.e.,
Northern pines and Southern pines) were proposed in Albani et al., 2006. Here, these PFTs are included and further
refined as seven major PFTs, namely early-successional broadleaf trees (EaSBT), middle-successional broadleaf
trees (MiSBT), late-successional broadleaf trees (LaSBT), northern and southern pines (NSP), late-successional
conifers (LaSC), C3 shrubs and grasses (C3ShG), and C4 shrubs and grasses (C4ShG) (Supplement S1). The
broadleaf PFTs (i.e., EaSBT, MiSBT, and LaSBT) are distinguished between tropical and non-tropical subtypes.
These PFTs primarily differ in their phenology, leaf physiological traits, allometry, mortality rate, and dispersal
distance. As in previous versions of ED, the spatial distribution of PFTs is mechanistically determined by individual
competition for light, water and nutrients. No quasi-equilibrium climate–vegetation relationships, or other
assumptions or observations, are used to constrain the presence or absence of PFTs.

Leaf physiology determines short-term (i.e., < hourly) leaf-level carbon and water exchanges in response to
environmental conditions (air temperature, shortwave radiation, air humidity, wind speed, $CO_2$ level). The
representation of leaf level physiology in previous versions of ED (Moorcroft et a., 2001) was taken from IBIS
(Foley et al., 1998), which in turn was based on prior work from Farquhar, Collatz, Ball and Berry and others





(Farquhar and Sharkey 1982; Ball et al., 1987; Collatz et al., 1991, 1992). Here, ED's representation of leaf level
physiology is reformulated for C3 and C4 pathways (Farquhar et al., 1980; Von Caemmerer and Furbank, 1999)
with added boundary layer conductance for diffusing water vapor and $CO_2$ between ambient air and leaf surface, and
parameterized with temperature dependence functions from other studies (Bernacchi et al., 2001; von Caemmerer et
al., 2009; Kattge and Knorr, 2007; Massad et al., 2007; Von Caemmerer, 2000, Supplement S3).

Hydrology controls the water available for vegetation. The hydrology submodule in ED tracks soil moisture
dynamics between incoming water flow from precipitation and outgoing flow through percolation, runoff, and
transpiration. Previous ED versions did not include evaporation from soil and canopy and also did not account for
snow dynamics. Here, evaporation from soil and canopy is estimated based on the Penman-Monteith (P-M) equation
(Monteith, 1965; Mu et al., 2011). In addition, a simple snow dynamics process is introduced to decrease water
availability for plants when air temperature drops below the freezing point and increase it when air temperature rises
above freezing point at a rate depending on air temperature. More details can be found in Supplement S9.

Land use activities (e.g., deforestation and wood harvesting) remove vegetation carbon from ecosystems for various
purposes. This carbon is traditionally tracked in wood product pools, with different lifetimes and temporal emissions
to the atmosphere. The previous land use submodule in ED only tracked changes in vegetation and soil carbon
during various land use activities but did not track subsequent decay process of product pools (Hurtt et al., 2002). In
ED v3.0, three wood product pools are added to track the life cycles of harvested wood and associated decay
processes (Supplement S11). Wood product pools gain carbon from land use activities such as wood harvesting or
deforestation, and lose carbon through decay and emissions to the atmosphere. The loading of these product pools,
and their decay rates, are based on a prior study (Hansis et al., 2015).
**2.2 Model initialization**
Global spin-up of ED initialized ecosystems to contemporary conditions by taking into account climate change,
rising $CO_2$, and land use change. The global spin-up was comprised of two separate runs at 0.5° spatial resolution.
The first run, called the "equilibrium simulation", ran ED from initial conditions to equilibrium. This run was
performed for 1000 years by which time PFT composition and carbon pools of vegetation and soil reached a
dynamic equilibrium. The second run, called "transient simulation", restarted from the end of the equilibrium
simulation and simulated for 1166 years, corresponding to the period A.D. 851 – A.D. 2016, with varying $CO_2$
levels, land-use change, and climate variability. Both runs were driven with meteorological forcing from NASA
Modern-Era Retrospective analysis for Research and Applications, version 2 (MERRA2) (Gelaro et al., 2017) and
surface $CO_2$ concentration from NOAA CarbonTracker Database, version 2016 (NOAA CT2016) (Peters et al.,
2007, with updates documented at http://carbontracker.noaa.gov). Additionally, the transient simulation run utilized
prescribed burned area from the Global Fire Emissions Database, version 4 (GFED4) (Randerson et al., 2015) and
forced land-use change from Land Use Harmonization, version 2 (LUH2) (Hurtt et al., 2019b, 2020). Details of
these simulations are provided below.




The equilibrium simulation was started from bare ground where the soil and vegetation carbon pools were set at
zero, and all PFTs were initialized with equal seedling density for all patches and all grid cells over the globe. This
run was driven for 1000 years with MERRA2 climatology of 1981-1990 and NOAA CT2016 average surface $CO_2$
between 2001-2014 (with spatial variation and global average rescaled to 280 ppm). No climatic envelope or
potential biome maps were used to constrain PFT spatial distribution; competition determined final PFT
distributions, vegetation structure, and carbon stocks. The land-use change module was disabled in this run of
simulation.

The transient simulation was restarted from equilibrium conditions. The land-use change submodule was activated,
and all land-use transition types from LUH2 were incorporated into the simulation at annual time steps. These
transitions included changes in agriculture and forest extent, shifting cultivation, and wood harvesting, among
others. MERRA2, NOAA CT2016 and GFED were used throughout the simulation with varying temporal settings
depending on data availability. Specifically, for MERRA2, a climatology between 1981-1990 was used until 1981,
and annual meteorology was used subsequently. For NOAA CT2016, an average surface $CO_2$ concentration between
2001-2014, which varies spatially and grows over time, was used until 2000, while annual NOAA CT2016 surface
$CO_2$ concentrations were used subsequently. For GFED4 burned area, an average between 1996-2016 was used until
1996, after which annual burned area was used.

**2.3 Forcing data**

Meteorological variables utilized from MERRA2 include surface air temperature (TLML), surface specific humidity
(QLML), precipitation (PRECTOTCORR), incident shortwave radiation (SWGDN), surface wind speed (SPEED),
and multi-layer soil temperature (TSOIL1-TSOIL3). Original estimates of surface air temperature, surface specific
humidity, incident shortwave radiation, and surface wind speed were averaged from daily hourly to monthly hourly
for each year between 1981 to 2016. The resulting annual monthly average of diurnal meteorological variables were
used to drive the leaf physiology submodule in ED. Hourly surface air temperature, precipitation, and soil
temperature were also aggregated to monthly averages for each year between 1981 to 2016, and then used to drive
the soil hydrology, phenology, evapotranspiration, and biogeochemical modules in ED.

Surface $CO_2$ concentration was extracted from the lowest vertical level of NOAA CT2016 $CO_2$ mole fraction which
is temporally and spatially varying. The original datasets were first linearly interpolated from 3°x2° (longitude x
latitude) to 0.5°x0.5° and from 3-hour to hourly, and then averaged to monthly hourly estimates for each grid cell
and each year between 2001 and 2014, resulting in surface $CO_2$ concentration maps with 4032 timesteps (14 years,
24 hours, 12 months) for each 0.5°x0.5° grid. The surface $CO_2$ concentration maps were used to drive the transient
simulation from 850 to 2000, retaining average spatial variation between 2001 and 2014 and applying a scaling
factor to force the global annual average $CO_2$ concentration to remain at 280 ppm before 1850, then grow linearly to


310 ppm in 1950 and to 375 ppm in 2000. This increasing trend in global average matches observed $CO_2$ growth
rates from Keeling (2008).

LUC forcing was derived from the LUH2 (version v2h) for years 850-2015 (Hurtt et al., 2019b, 2020). The original
land use state and land use transitions were aggregated from a spatial resolution of 0.25°x0.25° to 0.5°x0.5° for each
year between 850 and 2015. Subtypes of land use states and associated transitions were grouped into the major land
use types of the model's predecessor version (LUH1). Specifically, sub crop types of C3 annual crops (c3ann), C3
perennial crops (c3per), C4 annual crops (c4ann), C4 perennial crops (c4per) and C3 nitrogen-fixing crops (c3nfx)
were all merged as cropland. Forested primary land (primf) and non-forested primary land (primn) were merged as
primary land; forested secondary land (secdf) and non-forested secondary land were merged as secondary land; and
managed pasture (pastr) and rangeland were merged as pasture. Note that all types of land use transitions and gross
transition rate were used in ED's land use module.

Soil properties, including depth, hydraulic conductivity, and residual and saturated volumetric water content are
important for determining plant water availability. These soil properties were taken from Montzka et al. 2017.
Additional details can be found in the supplement (S9, hydrology submodule).
**2.4 Model evaluation**
A benchmarking package of data (Table 1) was collected to evaluate ED performance. Eight critical variables were
assessed in four categories including: PFT distribution, carbon stocks in vegetation and soil, carbon and water
fluxes, and vegetation structures in terms of canopy height and vertical LAI. Evaluation was carried out at different
spatial (grid, latitudinal, and biome) and temporal scales (climatological, seasonal, and interannual). For each
variable, a widely used dataset was used for reference, and in some cases, these span different years. An important
feature of our method was to adjust the simulation years from ED to match each benchmarking dataset.
**2.4.1 Vegetation distribution**
The satellite-based land cover product, ESA CCI, was used to examine the distribution of three modelled PFTs,
grass, broadleaf trees, and needleleaf trees (ESA 2017). Many satellite-based land cover datasets differ largely from
ED in PFT definition. For example, no successional PFTs exist in ESA CCI land cover types. Thus, the native PFTs
in ED and ESA CCI both have to be aggregated to broader categories such as broadleaf PFTs, needleleaf PFTs, and
grass PFTs. To do this, the 22 native land cover classes of ESA CCI were first reclassified to 'broadleaf evergreen
tree', 'broadleaf deciduous tree', 'needleleaf evergreen tree', 'needleleaf deciduous tree', 'natural grass' and
'manned grass' using a cross-walk table (Poulter et al., 2015). They were then further merged by phenology type
and aggregated to 0.5°, resulting in PFT fraction maps of broadleaf PFTs, needleleaf PFTs, and grass and shrub
PFTs. ED PFTs of EaSBT, MiSBT and LaSBT were merged as broadleaf PFTs, NSP and LaSC were merged as
needleleaf PFTs, and C3ShG and C4ShG were merged as grass and shrub PFTs.





**2.4.2 Carbon fluxes**
Evaluation of carbon fluxes focused on GPP and NBP. Modelled GPP was evaluated with respect to spatial pattern,
seasonality, and interannual variability using two satellite data-driven GPP datasets, FLUXCOM (Jung et al., 2020)
and FluxSat (Joiner et al., 2018), and the satellite-retrieved sun-induced chlorophyll fluorescence (CSIF) dataset
(Zhang et al., 2018). The FLUXCOM and FluxSat datasets are derived from a data-driven approach that combines
carbon fluxes measurements from FLUXNET and satellite observations from MODIS. Major differences between
FLUXCOM and FluxSat include the use of meteorological forcing and the specific approach used. FLUXCOM used
meteorological forcing and a machine learning approach, while FluxSat used a simplified light-use efficiency model
that does not rely upon meteorological forcing. FluxSat also used satellite-based sun-induced chlorophyll
fluorescence (SIF) to delineate highly productive regions. Satellite measurements of SIF have recently been
suggested as a promising proxy of terrestrial GPP, exhibiting high sensitivity to plant photosynthetic activities (Lee
et al., 2013; Guanter et al., 2014; Yang et al., 2015). In this study, we chose the CSIF dataset for its improved
spatiotemporal continuity. CSIF is generated by fusing Orbiting Carbon Observatory-2 (OCO-2)-retrieved SIF and
MODIS reflectance data using a machine learning approach. FLUXCOM, FluxSat, and CSIF were all resampled to
monthly estimates at 0.5x0.5 spatial resolution before the evaluation.

Modelled net biome productivity (NBP) was compared against multiple sources including estimates from process-
based models, atmospheric inversions, and the 2020 global carbon budget (GCB2020) (Friedlingstein et al., 2020).
For process-based models, 17 DGVMs reported in the GCB2020 were used to calculate the respective net land sink
by differencing land uptake and land use emissions estimates (i.e., $S_{LAND} - E_{LUC}$). For atmospheric inversions, three
systems are used, namely CarbonTracker Europe (CTE) (van der Laan-Luijkx et al., 2017), Jena CarboScope
(version s81oc) (Rödenbeck et al., 2008) and the Copernicus Atmosphere Monitoring Service (CAMS) (Chevallier
et al., 2005). The three inversions all derive surface carbon fluxes using atmospheric $CO_2$ measurements, prior
constraints on fluxes, and an uncertainty and atmospheric transport model, but vary with respect to the specific data,
prior constraints, and transport models used (Peylin et al., 2013). In the GCB2020, the residual terrestrial sink was
used, which was calculated as total emissions from fossil fuel and land use change minus the atmospheric $CO_2$
growth rate and ocean sink (i.e., $E_{FF} + E_{LUC} - G_{ATM} - S_{OCEAN}$).
**2.4.3 Carbon stocks**
Modelled carbon pools were evaluated with regards to vegetation aboveground biomass (AGB) and soil carbon. The
reference AGB data included estimates from Santoro et al. (2018) and Spawn et al. (2020). These two AGB datasets
provide high spatial resolution (e.g., 100 m to 1000 m) wall-to-wall global estimates of the year 2010, but differ in
their methodologies. Specifically, AGB from Santoro et al. (2018) was produced by combining spaceborne synthetic
aperture radar (SAR) (ALOS PLASAR, Envisat ASAR), Landsat-7, and Lidar observations from Ice, Cloud, and
land Elevation Satellite (ICESat). AGB from Spawn et al. (2020) includes biomass of forests and also other woody
non-forest plants. Reference soil carbon was from the Harmonized World Soil Database (HWSD) (Wieder et al.,
2014), including soil carbon for topsoil (0 to 30 cm) and subsoil (30 to 100 cm).



### 2.4.4 Vegetation structure


Evaluation of modelled forest structure focused on total and vertical distribution of leaf area index (LAI) and tree
canopy height. Two reference LAI products, namely MODIS MCD15A3H (Myneni et al., 2015) and GEOV2 LAI
(Verger et al., 2014), are used for evaluating total LAI in terms of spatial distribution, seasonality, and interannual
variability. The MODIS and GEOV2 LAI datasets were both derived from passive optical observations with
empirical-based inversion methods which relate leaf area with optical canopy reflectance or vegetation indices;
however, these two products vary with source of optical observations and choices for inversion methods. Reference
vertical LAI was from the Global Ecosystem Dynamics Investigation (GEDI) L2B products, which retrieves leaf
vertical distribution from lidar waveform return (Dubayah et al., 2020b). Reference canopy height data were based
on direct forest structure observations from GEDI L2A (Dubayah et al., 2020c) and the ICESat-2 ATL08 products
(Neuenschwander et al., 2020). Mean canopy height was generated at 0.5° spatial resolution from the relative height
98[th] percentile (RH98) of all GEDI L2A footprints and canopy top height (h_canopy) of all ICESat-2 ATL08
segments of good quality.

### 3 Results


ED results were evaluated across four primary categories: PFT distribution, vegetation and soil carbon pools, carbon
and water fluxes, and vegetation structure. Evaluation included comparing modelled global quantities, and their
associated spatial and temporal patterns, to the benchmarking datasets.

### 3.1 Evaluation of PFT distribution


Global total area of broadleaf PFTs, needleleaf PFTs and grass and shrub PFTs were estimated by ED to be 24.30,
8.93 and 24.63 million km$^2$ respectively. These results compare to ESA CCI data which estimate the same
respective global PFT areas at 20.13, 10.65 and 41.49 million km$^2$. The global spatial distribution and corresponding
zonal distribution of broadleaf PFTs, needleleaf PFTs and grass and shrub PFTs are shown in Fig. 3. In this
comparison, the major patterns of ED estimated PFT distribution were similar to the observed distribution of PFTs.
ED estimated needleleaf PFTs were dominate at high latitudes, broadleaf PFTs dominated in the tropics, and grass
and shrub PFTs were widespread globally. ED also predicted the observed coexistence of broadleaf and needleleaf
PFTs in southern China and eastern US. However, beyond these major patterns, ED estimates differed in some
specific regions. For example, ED predicted the existence of needleleaf PFTs along the Andes Mountains in South
America and in southern Australia. While this pattern was not evident in the ESA CCI data, there are other studies
based on ground observations that support it (Farjon and Filer, 2013). ED also estimated relatively more broadleaf
PFTs in eastern Europe and southern China, less broadleaf PFTs in Africa savanna, less needleleaf PFTs in east
Siberia, and less grass and shrub PFTs both in Africa savanna and northern China. Analogous results can also be
seen zonally, where major patterns of PFTs are broadly similar to observed but with some specific differences. In
terms of zonal distribution per PFT, the smallest discrepancies between ED and ESA CCI appear in broadleaf PFTs,
followed by needleleaf PFTs, and grass and shrub PFTs.



**3.2 Evaluation of AGB and soil carbon**


ED estimates of AGB were compared to corresponding benchmark data. ED estimated global total aboveground
vegetation carbon (including forest and non-forest) at 298 Pg C in 2010. This compares to 283 Pg C and 297 Pg C
estimated by Spawn et al. (2020) and Santoro et al. (2018). ED's estimate of the spatial pattern of AGB was also
comparable to that of both two benchmark datasets, with the highest biomass densities across the tropics (i.e., the
Amazon rainforest, the Congo river basin, and southeast Asia) with declining biomass densities northward towards
the temperate and boreal regions. For example, similar to observations, average estimated AGB density was ~15 kg
C/m$^2$ in the tropics and less than 2.5 kg C/m$^2$ across temperate and boreal regions (Fig. 4d). In addition, the AGB
transition along the African forests-savanna zone was represented by ED, albeit with lower values in the savanna.
Major discrepancies between ED and benchmarking data appear in southern China, southeast Asia and southeast
Brazil.

ED estimates of soil carbon were compared to benchmark data on soil carbon. ED estimated total global soil carbon
at 671 Pg C in 2000, which was within the range of CMIP5 ESMs (510 - 3040 Pg C) (Todd-Brown et al., 2013), but
lower than the HWSD estimate of 1201 Pg C. Comparing total stocks at the biome level (Fig. 5d) showed that ED
generally reproduced soil carbon variation across biomes, but notably underestimated carbon in boreal forest/taiga,
deserts and xeric shrublands, tropical and subtropical grasslands, savannas and shrubland. The soil carbon map from
ED revealed different spatial patterns compared to HWSD, with relatively less spatial heterogeneity and fewer
regions with densities above 30 kg C/m$^2$.

**3.3 Evaluation of GPP, NBP and ET**


Globally, the ED estimate of average annual GPP was 134 Pg C yr$^{-1}$ between 2001-2016, which compares to 120 Pg
C yr$^{-1}$ from FLUXCOM and 136 Pg C yr$^{-1}$ from FluxSat over the same period. The spatial pattern of GPP from ED
was also compared to benchmark values at the grid and latitudinal scales (Fig. 6). Similar to observations, areas of
highest productivity occur in the tropics, followed by temperate and boreal regions. For the tropics, ED was ~0.5 kg
C/m$^2$/yr higher than FLUXCOME, and ~0.2 kg/C/m$^2$ higher than FluxSat, but lower than both over the Africa
Savanna. Additionally, ED was relatively higher in southern China and Brazil than either benchmark dataset. A
notably increasing annual trend in total global GPP can be seen in both ED and FluxSat estimates between 2001-
2016 as well as from globally averaged CSIF (Fig. 7). ED also reproduced GPP interannual variability from FluxSat,
FLUXCOM and CSIF, dipping in the years 2005, 2012 and 2015, and peaking in 2006, 2011 and 2014. Regarding
latitudinal seasonality at the biome scale (Fig. 8), ED captured GPP timing for most latitudinal zones including 60° -
90°N, 45° - 60°N, 15° - 30°N and 60° - 30°S. Major differences appear in 30° - 45°N, where ED shows decreases
from July-September, and in 15°S - 0°, where ED shows delayed monthly timing of lowest annual GPP values.

Globally, the ED estimate of average annual NBP between 1981 and 2016 was 1.99 Pg C/yr, which can be
compared to 1.21-1.80 Pg C/yr from atmospheric inversions, 1.11 Pg C/yr from DGVMs, and 1.31 Pg C/yr from
GCB2020 residual terrestrial sink. ED estimates were also compared to benchmark datasets on global changes over



time (Fig. 9). Similar to the references, ED estimated an increasing trend with substantial interannual variation
during the 1981-2015 period. This variation included reductions in El Niño years (such as 1983, 1998 and 2015) and
increases in La Niña years (such as 1989, 2001-2002 and 2011). An exception is 1991-1992, where ED and DGVMs
were both lower than atmospheric inversions. This period includes the Mt. Pinatubo eruption the effect of which is
not included in the shortwave radiation forcing of GCB2020 DGVMs or ED (Mercado et al., 2009; Friedlingstein et
al., 2020). During the period 2007-2016, ED produced a continued increasing trend over the 2007-2016 period as
reflected in the mean of atmospheric inversions, but not the mean of DGVMs. Specifically, ED estimated NBP
averaged 2.34 Pg C/yr from 2007-2016, which as within the range of the atmospheric inversions estimates (1.77 -
2.64 Pg C/yr) and DGVMs estimates (0.58 - 2.82 Pg C/yr), but higher than either the mean of DGVMs (1.40 Pg
C/yr) or the GCB2020 residual terrestrial sink (1.81 Pg C/yr). Despite the similarities in global trends, the latitudinal
comparison between ED and atmospheric inversions indicated contrasting attribution of the global sink (Fig. 10). In
comparison to the atmospheric inversions, ED predicted a stronger sink in tropics and relatively weaker sink in the
Northern Hemisphere. Such a pattern was highlighted in the global carbon budget (Friedlingstein et al., 2020),
where process-based models and the atmospheric inversions generally show less agreement on the spatial pattern of
the carbon sink in these two regions. There is recognized uncertainty about the underlying actual pattern due in part
to the in-situ network, which is spatially biased towards the mid-latitudes (i.e., more observational sites) relative to
the tropics (i.e., fewer observational sites) (Ciais et al., 2014b).

Globally, the ED estimate of global mean annual ET between 1981 and 2014 was 393.46 mm/yr, which can be
compared to 582.10 mm/yr from FLUXCOM. ED estimates of ET were also compared to gridded FLUXCOM data
and by latitude (Fig. 11). Similar to the reference dataset, ED estimated the highest rates across the tropics with
decreases towards high latitudes. This pattern generally followed the spatial distribution of precipitation. ED estimates
were close to FLUXCOM over the tropics (i.e., 1500 mm/yr) as well as latitudes above 60°N and below 35°S (i.e.,
below 500 mm/yr), but notably underestimated average annual ET in other latitudes. ED estimates were generally
smaller than FLUXCOME in dry regions such as southern Africa and interior Australia.
**3.4 Evaluation of canopy height and LAI vertical profile**
Evaluation of vegetation structure estimates focused on leaf area and canopy height. Fig. 12 presents the spatial
distribution of growing season LAI from ED, GEOV2, and MODIS. Growing season LAI is chosen for comparison
because winter snow in the northern region (e.g., boreal forests) might affect LAI retrieval and cause uncertainties in
remote sensing estimates (Murray-Tortarolo et al., 2013). There was good agreement in spatial pattern between ED
and reference LAIs (Fig. 12d), showing peaks in the tropics and boreal region (near 50°N), and relatively low
estimates across temperate regions. In the tropics, ED estimated an average LAI of 6.0 $m^2/m^2$, which was similar to
GEOV2 but higher than MODIS. However, ED produced higher LAI in temperate and boreal regions than both
reference datasets, specifically in southern China and Brazil. Despite these differences there was a general
agreement in the greening trend between 1999 and 2016 (as shown in Fig. 13). The linear fitted LAI trend was 0.058
$m^2/m^2$ per decade for ED, 0.090 $m^2/m^2$ for GEOV2. and 0.046 $m^2/m^2$ for MODIS. LAI seasonality was also





compared across latitudinal bands in Fig. 14. Similar to references, ED captured peak season in latitudinal bands 60°
- 90°N, 45° - 60°N, and 60° - 30°S, but shows less agreement with the references in the tropics (0° - 15°N and 15S°
- 0°). In addition, ED LAI in winter is larger than either reference LAI; at latitudes above 45°N, and between 30°N
and 45°N, ED LAI is higher for all seasons. Similarly, higher LAI also appears in 60°S - 30°S, across southern
China and Brazil.

The estimated vertical profile of LAI from ED was compared to GEDI both spatially and by latitude band. Spatially,
ED and GEDI L2B had a similar spatial pattern with most vegetated regions having concentrated LAI under 10m,
and only tropical forests, part of southern China and the US having substantial LAI above 30m (Fig. 15).
Comparisons of LAI profiles by latitude band indicate close agreement in each zone, and with all regions having the
highest values of LAI closest to the ground (0-5 m) and decreasing with canopy height (Fig. 16). Discrepancies can
be seen at the 0 - 5m and 10 - 15m LAI interval along most latitudinal bands, where ED tends to be higher 0-5m,
and lower in the 10-15m bin.

Tree canopy height estimates from ED were compared with satellite lidar observations from GEDI and ICESat-2
(Fig. 17). Like the reference datasets, ED produced a spatial pattern with taller trees in tropical rainforests, southern
China and the eastern US. The canopy height gradient from forests to savannas in South America (northwest to
southeast) and in Africa (central to north and south) were also generally captured by ED. Latitudinal comparison
shows ED estimated average height is above 30 m in tropics and is ~10m in temperate regions. The general
differences between ED and reference datasets are less than 10m across all latitudes. However, ED tree height in
southern China and Brazil was higher than the references, and lower than references across African savanna.
**4 Discussion and Conclusions**
In this study, we developed a new global version of the Ecosystem Demography model and evaluated it against
benchmark datasets for a wide range of important variables spanning carbon stocks, carbon and water fluxes,
vegetation distribution, and vegetation structure. Historically, different models have been developed separately in
areas of biogeochemistry, biogeography, and biophysics, and in some cases important patterns have been set through
observations or other prior constraints (Bonan, 1994; Dickinson, 1993; Haxeltine and Prentice, 1996; Hurtt et al.,
1998, Lieth, 1975; Neilson, 1995; Parton, 1996; Potter et al., 1993; Prentice et al., 1992; Raich et al., 1991; Sellers et
al., 1986). The ability of this model to reliably simulate such a wide range of phenomena globally in a single
mechanistic and consistent framework represents an important interdisciplinary synthesis, a functional modelling
advance, and to our knowledge is unprecedented.

ED estimation of carbon stocks and fluxes compared favourably to benchmarking datasets across a range of spatial
and temporal scales, from grid cell to global, from seasonal to decadal. Similar to benchmarking datasets, ED
produced latitudinal gradients of GPP and AGB, positive trend in global total GPP, global total AGB and GPP
within reference range, and interannual variation of NBP in response to El Niño and La Niña events. Producing such



patterns of both global carbon fluxes and stocks is challenging, as it requires models to have the ability to
mechanistically scaling up physiological processes from leaf to ecosystem scales. It also requires models to
accurately characterize responses of ecosystem demographic processes to climate change, soil conditions, and land
use activities. As a part of a new generation of DGVMs attempting to meet these challenges, ED leverages advances
in detailed understanding of ecosystem-physiology (e.g., Ball–Berry stomatal conductance model and Farquhar
photosynthesis model) (Ball et al., 1987; Farquhar 1980), soil biogeochemistry (e.g., CENTURY soil model) (Parton
1996), and processes of disturbance and recovery (e.g., LUH1/LUH2 modelling of land-use transition through time)
(Hurtt et al., 2011, 2020). This study is the first, to our knowledge, to combine ecosystem demography and land use
history to simulate global carbon dynamics and compare to a wide range of benchmarks.

In addition to carbon stocks and fluxes, ED simultaneously estimated the spatial distribution of the major PFTs
globally. ED produced dominance of broadleaf PFT in tropics and needleleaf PFT in high latitudes, which is similar
to benchmarking data. The ability to estimate these patterns mechanistically required the ability to characterize
functional plant traits and trade-offs of vegetation as well as the processes and timescales of competition for light,
water, and other resources. Numerous studies have made previous advances which contributed to the progress in this
study. For example, plant traits have been observed and compiled crossing a wide range species and geographical
domain (Reich 1997 et al., 1997; Kattge et a., 2011, 2020). Individual based/gap models have been developed to
track the life cycle of each individual tree and competition between individuals on plot and site level (Botkin et al.,
1972; Shugart and West 1977; Shugart et al., 2018; Pacala et al., 1996). Meanwhile, the SAS scaling approach was
previously developed to efficiently scale up the individual scale to ecosystem dynamics at regional and continental
scales (Hurtt et al., 1998; Moorcroft et al., 2001).

ED estimation of vegetation structure was also evaluated against benchmark data, in this case, novel observations
using lidar remote sensing. Impressively, ED mechanistically and independently produced latitudinal mean height
and LAI profiles similar to benchmarking datasets on vegetation structure. This progress is perhaps the most novel
achievement because progress on this topic was previously limited due to lack of global observations of vegetation
structure. Importantly, the ED model is natively height-structured, in that all trees have explicit height. Originally,
this feature was included to enable simulation of individual-based competition for light. This feature however also
offers the potential for direct connection to lidar observations on vegetation structure for the purpose of model
validation and/or initialization. Numerous studies have been completed at local and regional scales by initializing
the ED model with airborne lidar data, demonstrating the power of lidar technique in improving characterization of
contemporary ecosystems conditions (Hurtt et al., 2004, 2010, 2016, 2019a and Ma et al., 2021). The advent of
GEDI (Dubayah et al., 2020a) and ICESat-2 (Markus et al., 2017) has now expanded the potential for model
evaluation and initialization to global scales.

Despite all of these advances, no model is perfect. There are important examples of differences between ED
estimates and reference values that present important challenges for the future. Two examples are important to



consider further. First, ED estimates of AGB/GPP exceeded reference values in some regions, most notably southern
China, southeast Asia and southeast Brazil. Correspondingly, ED also tended to overestimate tree height in these
same regions. The discrepancies share a similar spatial pattern and are likely interrelated. One hypothesis is that this
overestimation may result at least in part from the land-use forcing. The LUH2 has been shown to underestimate
harvesting area on primary forest in southern China, and Southeast Asia for the period after 1950, and also
underestimates total cropland area in Brazil (Chini et al., 2021). LUH2 is being continuously updated and improved
through the contribution to the Global Carbon Budget project (Chini et al., 2021). Second, while relative patterns for
soil carbon showed close agreement at biome level for the majority of biomes, the absolute magnitude of soil carbon
was much lower than reference for several biomes and thus globally. Before over-interpreting these differences, it
should be noted that there are substantial uncertainties with current empirical soil carbon maps in terms of both
global totals and spatial distribution (Todd-Brown et al., 2013). Model errors in soil carbon may arise from poor
representations of biophysical conditions, inaccurate parameterization, or lack of other important drivers. Soil
carbon representation in ED, like that of many other DGVMs/ESMs, is highly simplified and strongly driven by
NPP and soil temperature, but these two drivers have been shown to only explain a small amount of spatial variation
in the HWSD map (Todd-Brown et al., 2013). The underestimation in the boreal forest/Taiga biome in particular is
likely due to a combination of these factors. Ongoing research through the NASA Arctic-Boreal Vulnerability
Experiment (ABoVE) program and the Next-Generation Ecosystem Experiments (NGEE Arctic) program will likely
improve both the data and model parameterization in this critical region.

Previous studies have developed benchmarking packages and designed model intercomparison activities to evaluate
model performance (Abramowitz et al., 2012; Collier et al., 2018; Eyring et al., 2016; Ghimire et al., 2016; Kelly et
al., 2013; Luo et al., 2012; Randerson et al., 2009; Sitch et al., 2008). Like those studies, we evaluated ED model
results using many key datasets and variables. The work here has utilized a particularly wide range of variables,
utilized the latest versions of key forcing data on climate and land-use, and added a new focus on vegetation
structure.

Future work will focus on addressing the limitations discussed above and making direct connections with lidar forest
structure observations from GEDI and ICESat-2 to improve demographic processes, and the quantification and
attribution of the terrestrial carbon cycle. Meanwhile, the global development and evaluation of ED demonstrates the
model's ability to characterize essential aspects of terrestrial vegetation dynamics and the carbon cycle for a range of
important applications. This model has recently been integrated with NASA's Goddard Earth Observing System,
Version 5 (GEOS-5) to forecast seasonal biosphere-atmosphere $CO_2$ fluxes in 2015-16 El Niño (Ott et al., 2018),
and used in NASA Carbon Monitoring System as the tool for high spatial resolution (e.g., 90 m) regional forest
carbon modelling and monitoring (Hurtt et al., 2019a; Ma et al., 2021), and also by NASA Global Ecosystem
Dynamics Investigation mission for quantification of land carbon sequestration potential (Dubayah et al., 2020a; Ma
et al., 2020). Results from these studies will likely be of importance to science applications, and also be used to
inform and prioritize future model advances. Meanwhile, the increasing number of remote sensing missions and



related data sets, advances in computation, and growing stakeholder interests in carbon and climate, as evidenced by
the Paris Agreement, bode well for future advances.

*Code and data availability.* All model simulation and source script can be found in
https://doi.org/10.5281/zenodo.5236771. All benchmarking datasets are cited and publicly available.

*Author contributions.* LM, GH, JF, SF and RS developed model code. LM, GH and LO designed this study. LM
conducted model simulation and evaluation. LM, GH and RL wrote main body of the manuscript. All authors
contributed to analysis and manuscript preparation.

*Competing interests.* The authors declare that they have no conflict of interest.

*Acknowledgements.* This work was funded by NASA-CMS (grant no. 80NSSC17K0710, 80NSSC20K0006, and
80NSSC21K1059), and NASA-IDS (grant no. 80NSSC17K0348).

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





 **Figures and Tables**

 Table 1. Summary of benchmarking datasets used for evaluation of ED model.

| Variable | Source | Description | Reference |
|---|---|---|---|
| Vegetation distribution | | | |
| PFT | ESA CCI | Global gridded, 300-m, 2015 | ESA (2017) |
| Carbon stocks | | | |
| AGB | Santoro et al. (2018) | Global gridded, 100-m, 2010 | Santoro et al. (2018) |
| | Spawn et al. (2020) | Global gridded, 300-m, 2010 | Spawn et al. (2020) |
| Soil carbon | HWSD | Gridded, 0.05 degree, 2000 | Wieder et al. (2014) |
| Carbon and water fluxes | | | |
| GPP | FLUXCOM (RS+METEO, CRUJIA and ERA5) | Global gridded, 0.0833-degree, 1979-2017 monthly | Jung et al. (2020) |
| | FluxSat | Global gridded, 0.05-degree, 2001-2018 monthly | Joiner et al. (2018) |
| NBP | CAMS (v17r1) | Global gridded, 1.875x3.75-degree, 1979-2017 monthly | Chevallier et al. (2005) |
| | Jena CarbonScope (s81oc_v2020) | Global gridded, 2.5x2.0 degree, 1981-2016 daily | Rödenbeck et al. (2008) |
| | CarbonTracker Europe (CTE) | Global gridded, 1x1 degree, 2000-2016 monthly | van der Laan-Luijkx et al. (2017) |
| | GCB2020 DGVMs | Global total, 1959-2019 yearly | Friedlingstein et al. (2020) |
| | GCB2020 Residual sink | Global total, 1959-2019 yearly | Friedlingstein et al. (2020) |
| ET | FLUXCOM (RS+METEO, CRUNCEP and GSWP3) | Global gridded, 0.0833-degree, 1981-2014 monthly | Jung et al. (2020) |
| Vegetation structure | | | |
| Tree height | GEDI L2A (v001) | 51°N ~ 51°S, 20-m footprint, 2019-2020 | Dubayah et al. (2020c) |
| | ICESat-2 ATL08 (v003) | 51°N ~ 51°S, 100-m footprint, 2018-2020 | Neuenschwander et al. (2020) |
| LAI | MODIS MCD15A3H (v006) | Global gridded, 500-m, 2003-2016 4-day | Myneni et al. (2015) |
| | GEOV2 | Global gridded, 1/3-km, 1999-2016 10-day | Verger et al. (2014) |
| Vertical LAI | GEDI L2B (v001) | 51°N ~ 51°S, 20-m footprint, 2019-2020 | Dubayah et al. (2020b) |



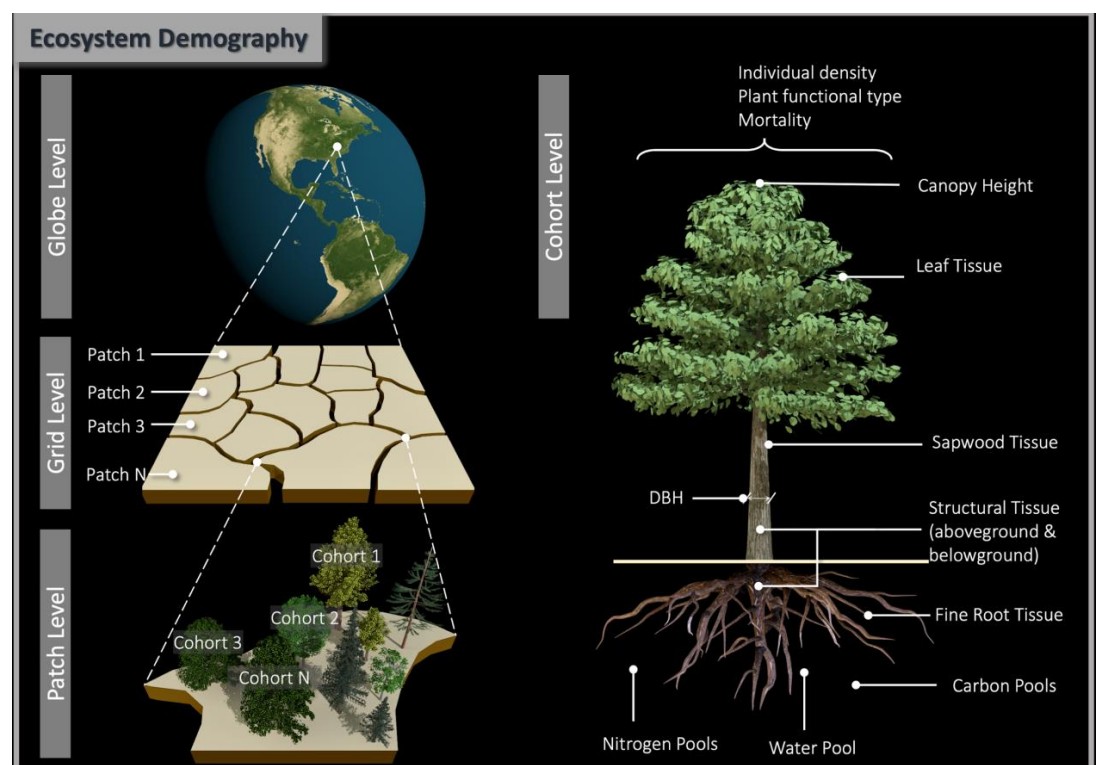

Figure 1. Diagram of vegetation representation scheme in ED model. Globe consists of land grids with fixed spatial resolution. A grid consists of patches with different ages from last disturbance and land use types, and patch areas dynamically change over time as a result of disturbance and land use changes. A patch consists of consists with different plant functional types and sizes. Plants in a cohort are depicted by properties including individual density, canopy height, diameter at breast (DBH), and biomass in leaf, sapwood, structural tissue and fine roots, and all these properties are simulated as a result of interaction with environment and other cohorts. Note that not all properties are shown here.

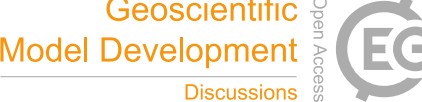



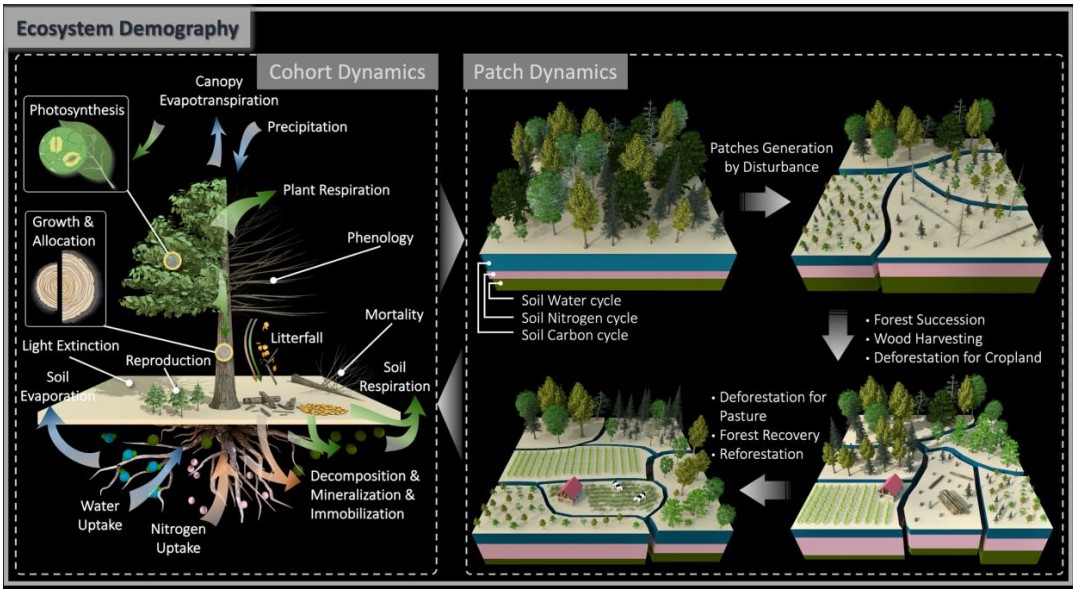

Figure 2. Schematic diagram of processes represented in ED model. Dynamics at cohort level consists of carbon-related flow (green arrow), water-related flow (blue arrow) and nitrogen-related (orange arrow). Carbon dynamics include carbon assimilation by photosynthesis, carbon allocation for plant growth in height/DBH, reproduction and respiration, carbon translocation between plants and soil through tissue turnover as litterfall and dead plants due to mortality, and carbon decomposition and respiration in soil carbon pools. Water dynamics include water inputs from precipitation and infiltration into soil, uptake by vegetation and evaporation and transpiration of soil and canopy. Nitrogen dynamics includes nitrogen uptake from soil pools, translocation from vegetation to soil through litterfall and dead plants, and mineralization and immobilization in soil. Note that not all processes that ED characterize are depicted here. Dynamics at patch level consist of consequences from a variety of disturbance events both natural and anthropogenic. Patch dynamics include disturbance-driven patch heterogenization in age and areas, forest succession, wood harvesting, deforestation for cropland and pasture expansion, and forest recovery and reforestation from abandoned cropland, harvested forest and pasture.



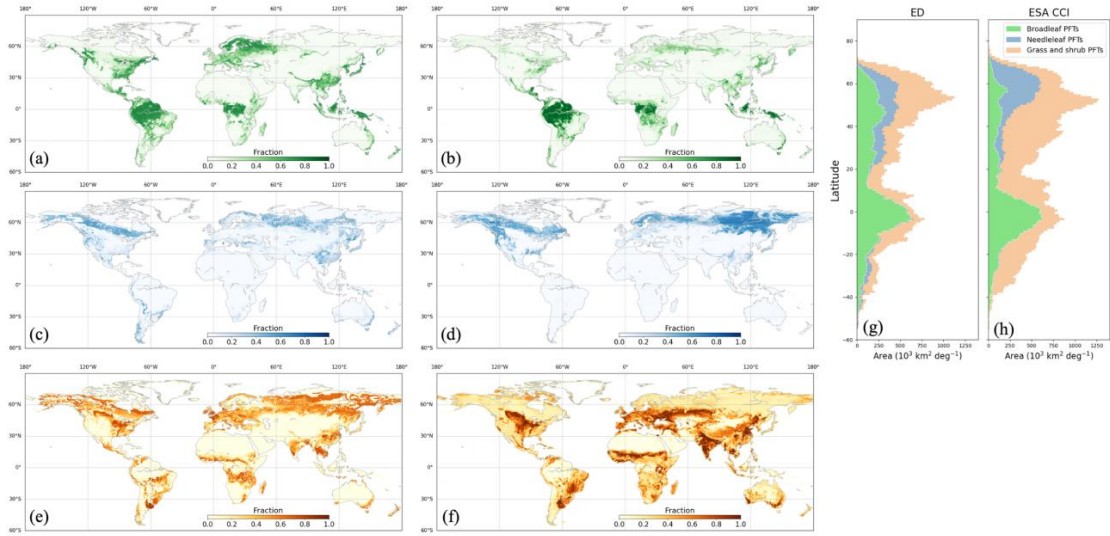

Figure 3. Spatial distribution of broadleaf PFTs, needleleaf and PFTs and grass and shrub PFTs in 2015 from ED (a),
(c) and (e), and from ESA CCI (b), (d) and (f). Corresponding latitudinal total area is compared in (g) and (h).



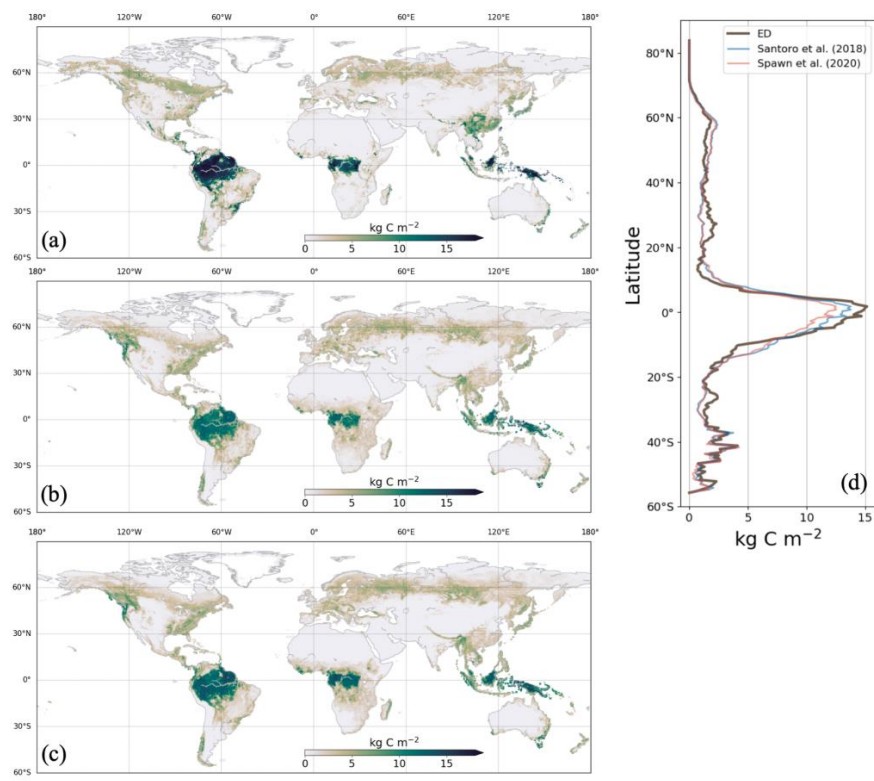

Figure 4. AGB in 2010 from ED (a), Spawn et al., (2020) (b), and Santoro et al., (2018) (c), with latitudinal average AGB compared in (d).





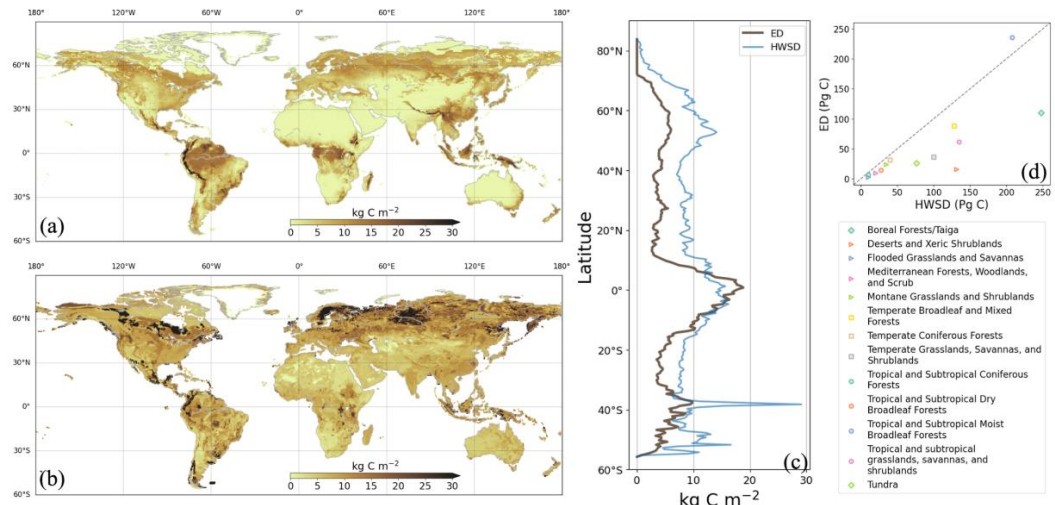

Figure 5. Soil carbon density in 2000 from ED (a) and HWSD (b). Latitudinal average density and total stocks per biome are compared in (c) and (d), respectively.



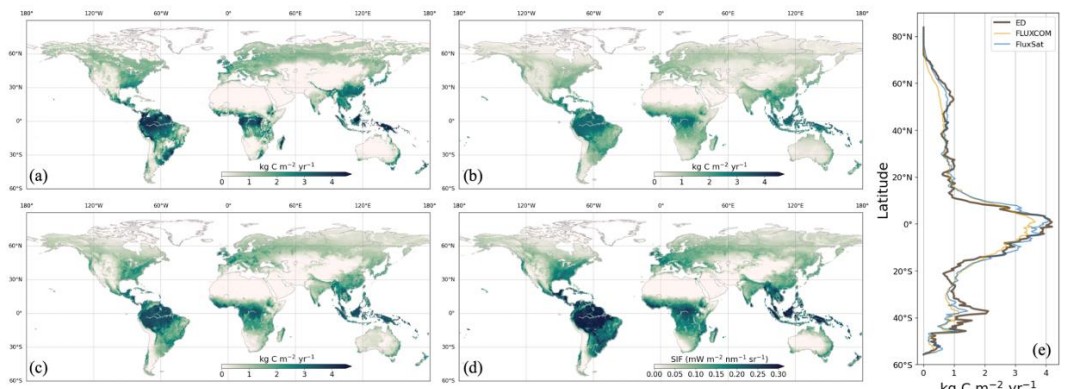

Figure 6. Average annual GPP between 2001 and 2016 from ED (a), FLUXCOM (b), FluxSat (c) and CSIF (d). Comparison of latitudinal average GPP is shown in (e).





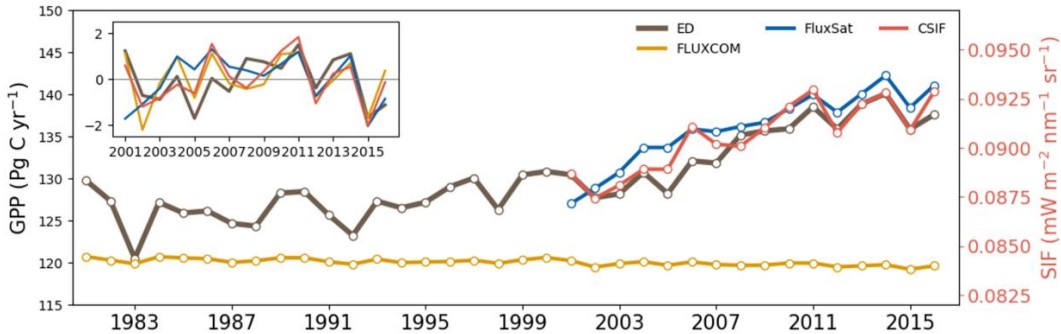

Figure 7. Time-series of global annual total GPP from ED, FLUXCOM, and FluxSat, and global annual average CSIF. Their interannual anomaly is shown in the inset.

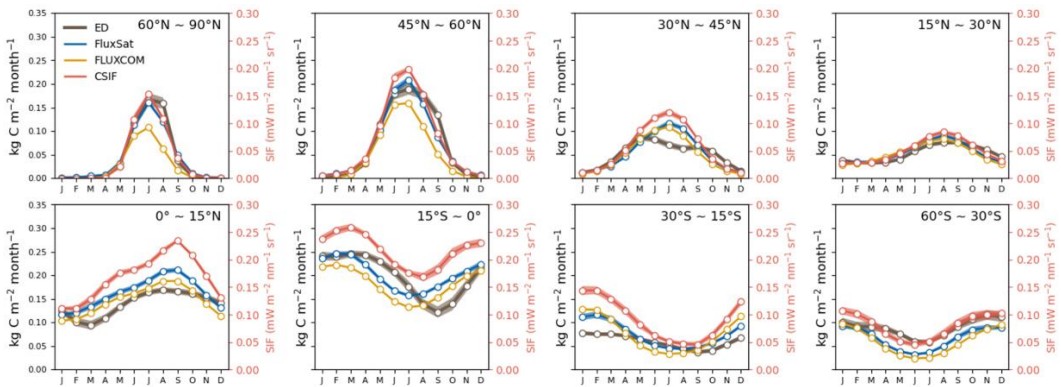

Figure 8. Average seasonal cycle (2001-2016) of GPP from ED, FLUXCOM, FluxSat, and CSIF by latitudinal band.



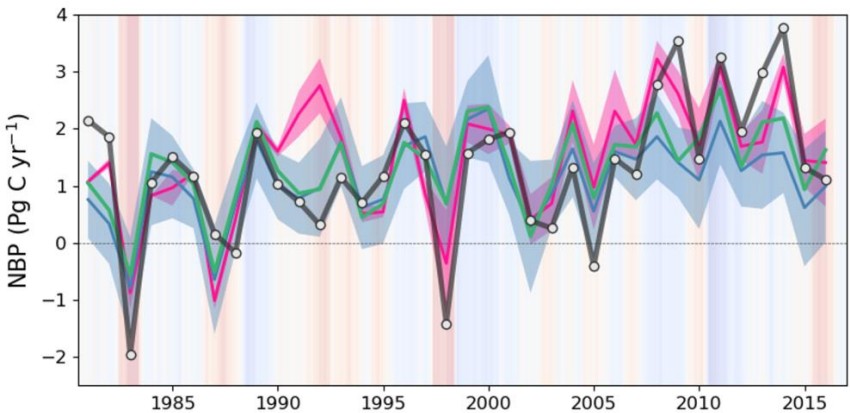

Figure 9. Global annual NBP between 1981 and 2016 from ED (black line), DGVMs from the GCB2020 (ensemble average shown in blue line with ±1σ spread shown in blue shading), the ensemble of atmospheric inversions (ensemble average shown in pink line with ±1σ spread shown in pink shading), and the terrestrial residual sink of the GCB2020 (green line). Positive values indicate net carbon uptake from land. Background shading represents the bi-monthly Multivariate El Niño/Southern Oscillation (ENSO) index, where red indicates El Niño and blue indicates La Niña.





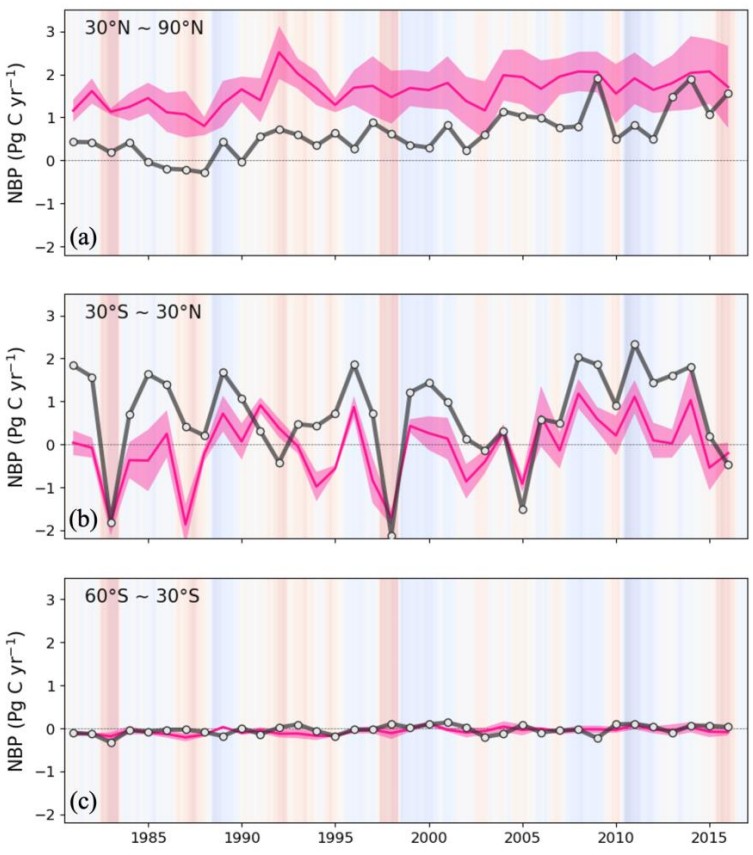

Figure 10. Annual NBP between 1981 and 2016 from ED and ensemble of atmospheric inversions for the Northern Hemisphere (>30°N) (a), tropics (30°N - 30°S) (b) and the Southern Hemisphere (<30°S) (c). Black line is ED, and the pink line and pink shading are the inversion ensemble average and ±1σ spread of atmospheric inversions, respectively.



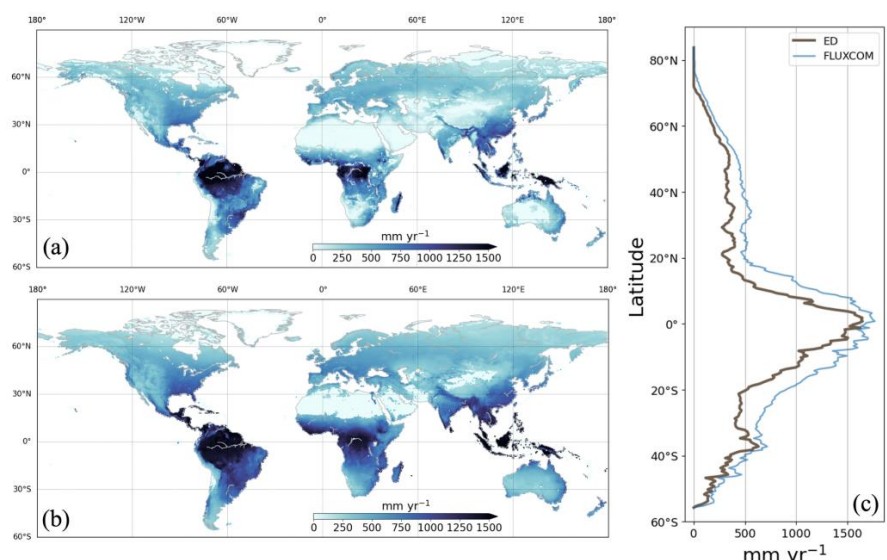

Figure 11. Average annual ET between 1981 and 2016 from ED (a) and FLUXCOM (b) with corresponding latitudinal average comparison (c).



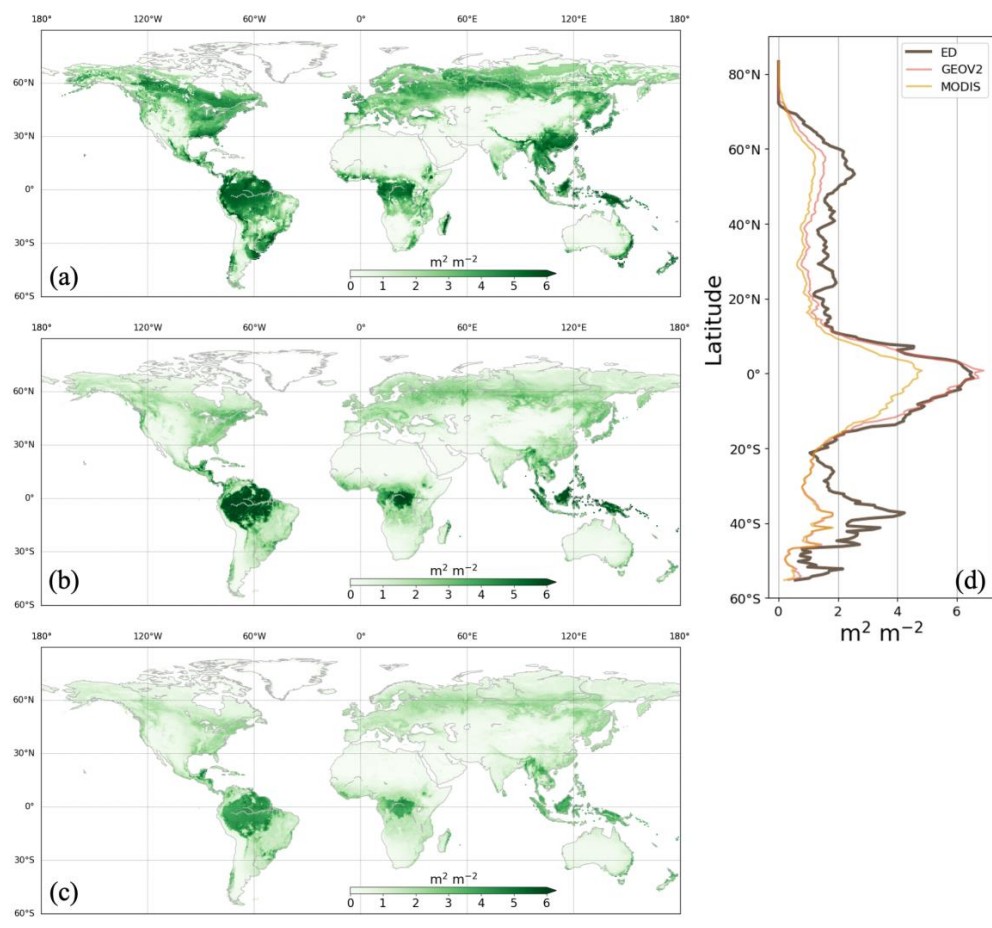

Figure 12. Average LAI during the growing season between 2003 and 2016 from ED (a), GEOV2 (b), and MODIS (c). Corresponding latitudinal averages are compared in (e). Growing season is defined as the months during which the average air temperature of MERRA2 is above 0°C.





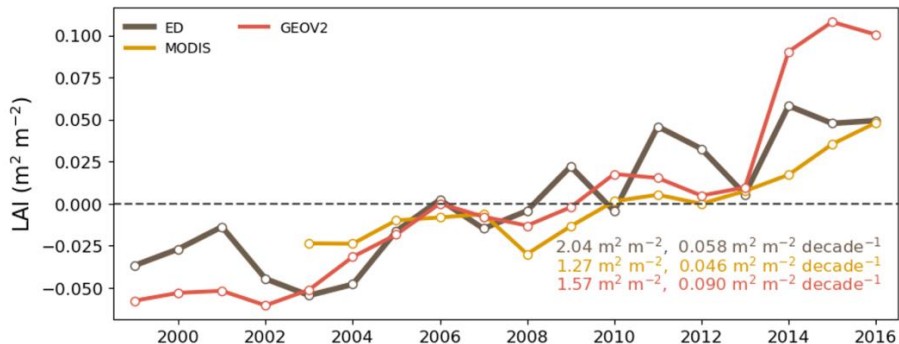

Figure 13. Interannual global average growing season LAI from ED, MODIS and GEOV2. The anomaly is calculated by subtracting annual LAI by multi-year average.





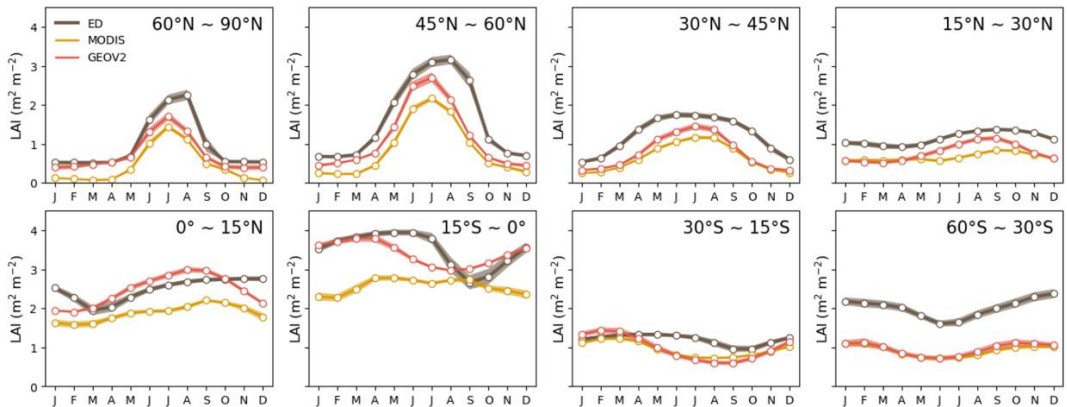

Figure 14. Seasonal LAI by latitudinal band from ED, MODIS and GEOV2.



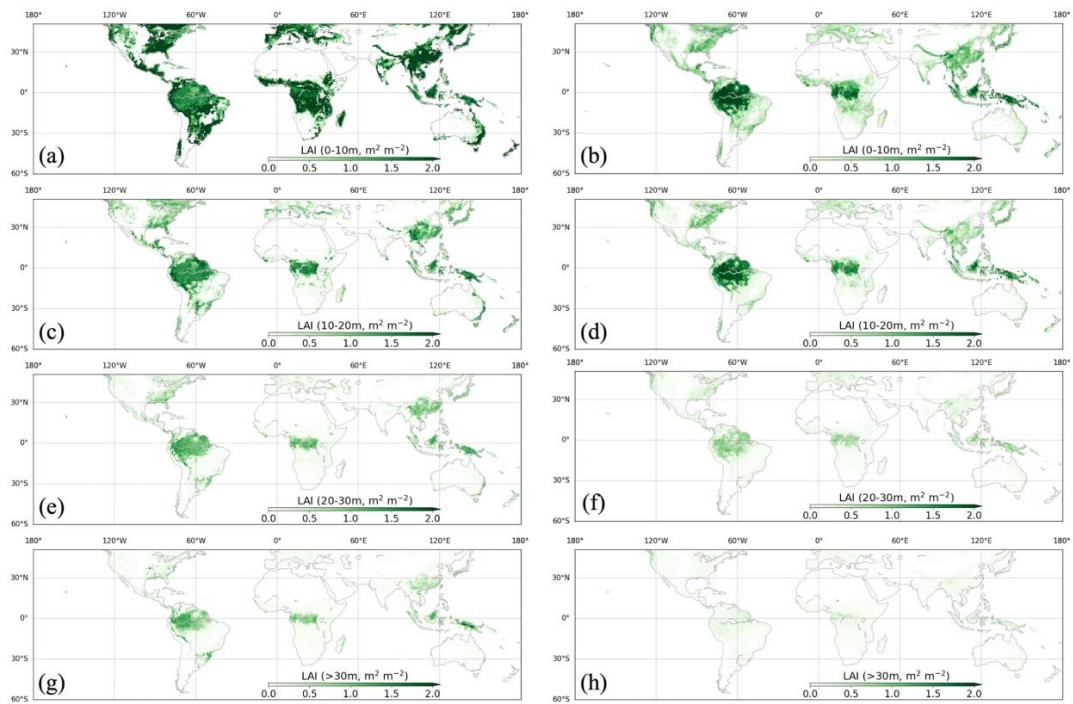

Figure 15. Vertical LAI from ED (left column) and GEDI L2B (right column) at height (0-10m) in (a) and (b), 10-20m in (c) and (d), 20-30m in (e) and (f), and above 30m in (g) and (h), respectively.



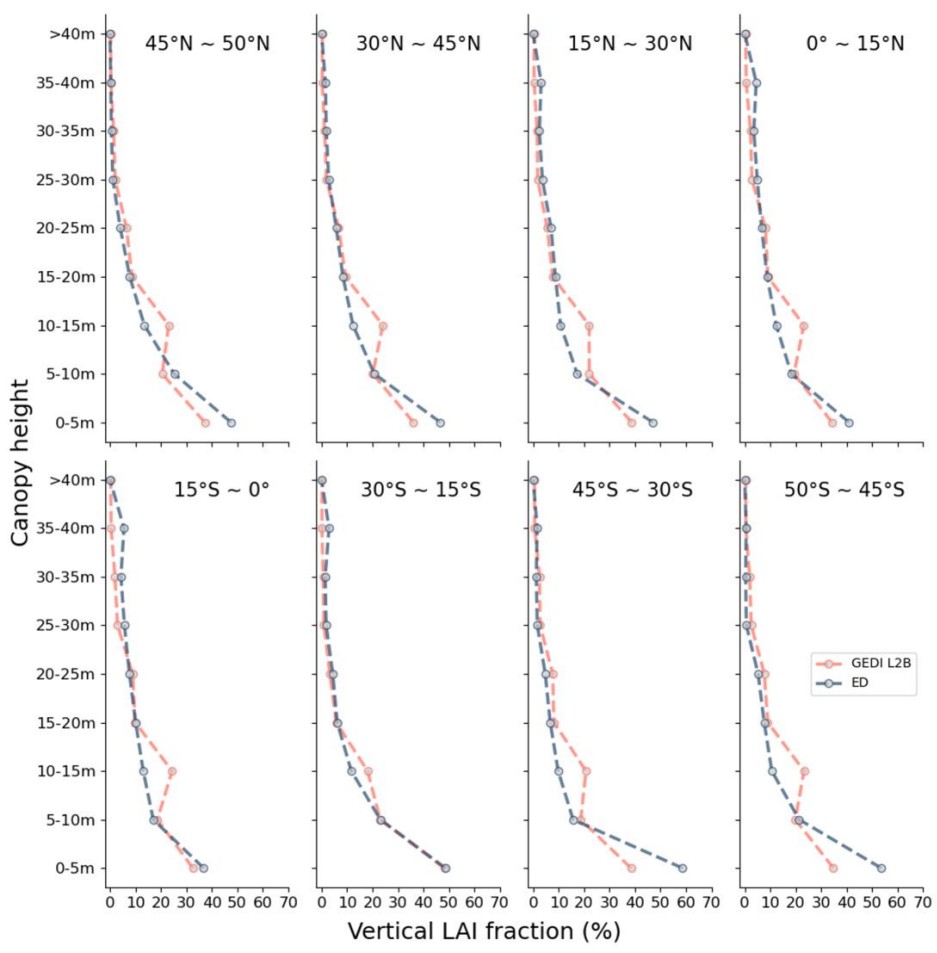

Figure 16. Relative fraction of vertical LAI by latitudinal band between ED and GEDI L2B.





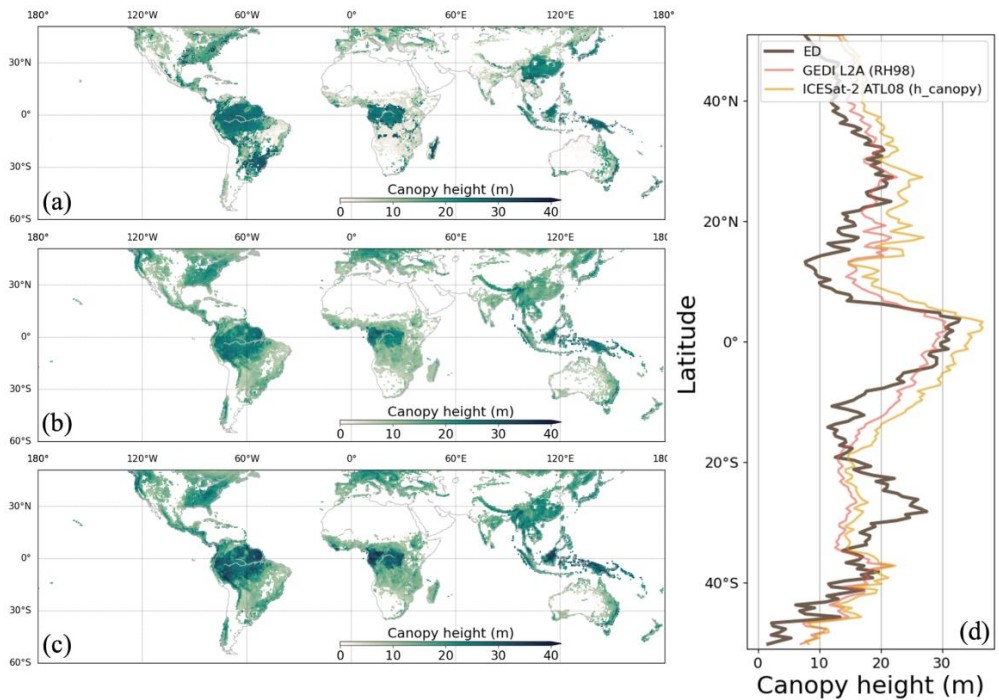

Figure 17. Canopy height from ED (a), GEDI L2A (b), and ICESat-2 ATL08 (c). Latitudinal averages are compared in (d). ESA CCI data grids with tree fractions below 5% are masked.