# Peer review of "Global Evaluation of the Ecosystem Demography Model (ED"

_Geoscientific Model Development, 2021_

## Author Comment (AC1)

Responses to reviews for manuscript **Global Evaluation of the Ecosystem Demography Model (ED v3.0)**

**To Reviewer #1:**

The manuscript compared global evaluation of water and carbon fluxes, vegetation dynamics such as leaf area index and canopy height reasonably to a series of global observations shown in table 1. As a whole, the manuscript seems carefully prepared and concise, and is thus suitable for the publication in GMD. At present, The improved ED-v3.0 model is a potential used as one of benchmark models for evaluating globally ecosystem responses to the climatic variations in the future. Using the model simulations, the authors showed spatio-temporal variations in GPP and AGB that were comparable to global observations well, including the effects of eccentric atmosphere-oceanic events like El nino/La nina on terrestrial carbon productivity. However, I suppose, in Discussion and Conclusions, you need to somehow modify the manuscript to more clearly show the importance and advantage of use of four modifications that has been never handled into the previous developed ED model. In particularly, the discrepancy between the present model and observations for evaluating global annual evaporation shown might be related to a lack of unknown terrestrial hydrological processes. Additional simulations were no longer required, while more discussions are to be indicated if the application of other new submodules also derived realistic products compared to those from the original ED. The manuscript will be thus accepted after covered moderately with these minor revisions. I will give several bullet points below.

**Response:** Thank you very much for your constructive comments which have improved the manuscript. We have revised the manuscript accordingly. Specifically, we added a new paragraph to the Discussion section to point out the qualitative benefits of the four key modifications (page 13, line 420-429). We also added more discussion on the potential causes of low ET (page 14, line 485-489).

Method line245: Make clear net biome productivity (NBP) here, and delete this in line260.

**Response:** We have revised it.

Results line 368-374: ED-evaluated annual evapotranspiration seems consistently smaller than FLUXCOM observations not only in dry regions you mentioned but across all the latitudinal ranges. Could you describe more detail what mechanisms regulated annual values in the model to explain the discrepancy between model and observational estimations. Additionally, how did the new hydrology submodule incorporated operate for the annual estimation associating with evaporation from soil and canopy? Or, Is this due to evaluations from Penman-Monteith big-leaf model? If so, for instance, Bonan et al.(2021) in Agri.For.Meteorol may help more discussion of the model work.

**Response:** This paper introduced evaporation estimation to the hydrology submodule, which is missing in previous regional versions of ED. In the Hydrology submodule, the ET is the sum of evaporation from soil, evaporation from canopy, and transpiration from canopy. The evaporation calculation follows the Penman-Monteith equations (Mu et al 2011) with further detail provided in the Supplement S9.

We agree that the ET underestimation appears not only in dry land but also other areas. There may be several causes of ET underestimation, but these were not explicitly identified in this study. We have added some discussion of these potential causes on page 14, line 485-489:

*... Third, ED estimates of ET were lower than reference across all latitudes. One reason for this difference could be the parameterization of Penman-Monteith equations in the Hydrology submodule, as the value of aerodynamic resistance used in this study was higher than reported in Mu et al 2011. A second potential cause could be the scaling of evapotranspiration (Bonan et al 2021), which combines cohort scale transpiration with patch scale evaporation and currently omits vertical variation of evaporation ...*

Discussion and Conclusions: line 408-417: Make the discussion started first. Move the first paragraph to the late part of this chapter and arrange then.

**Response:** We have revised this paragraph as suggested.

line 457: Delete the first sentence.

**Response:** Agreed. It has been revised.

line 459-465: The author stated here ED overestimated tree height in three particular regions like S.China, SE asia and SE Brazil. From Fig.17, the canopy height estimation from ED seems to be so smaller than observations over northern hemisphere, including the former two regions. Rather, ED-derived LAI from Fig.12 is larger over whole latitudes.

**Response:** ED-estimated height is indeed higher than GEDI over S. China and SE Asia; however, there is also underestimation in other areas at the same latitudes such as in Myanmar and Tanzania. Thus, this underestimation leads to overall lower height at these latitudes as shown in the Fig. 17d. Additionally, LAI in Fig.12 includes a contribution from grass shrub type of PFTs which lowers canopy height significantly.

line 478-483: Move these to the top of this chapter and arrange then.

**Response:** We have revised the Discussion section as suggested.

Other: Figure 5: In (c) and (d), symbols and captions in legend are too small.

**Response:** Agreed. We have redesigned it.

---

## Author Comment (AC2)

Responses to reviews for manuscript **Global Evaluation of the Ecosystem Demography Model (ED v3.0)**

**To Reviewer #2:**

Ma et al. evaluate an updated version of a land ecosystem model, the Ecosystem Demography model v3, which is a DGVM based on individual-based approach. In this paper, they first introduce the new model structure briefly, and then compared the simulated results, mainly on global carbon cycle, with reference datasets. The comparison includes global cumulative/average values, and spatial and temporal variations of carbon fluxes/pools. One of new features of the evaluation process is utilization of global observation on vegetation structure (vertical profile of LAI and canopy height), which can highlight the characteristic of the new ED model compared with other DGVMs. The evaluation revealed that the global carbon cycle processes were reproduced reasonably by the model, although the soil carbon distribution and its global value are still under debated. Overall, this paper is well structured and documented, and thus I judge this paper can contribute to the journal after minor revisions suggested below.

**Response:** Thank you very much for your constructive comments which have improved the manuscript. We have revised the manuscript accordingly. Specifically, we added more discussion on the soil carbon and tropical seasonality of GPP and LAI. We have also updated the evaluation regarding canopy height and vertical LAI as GEDI and ICESast-2 have released new versions of canopy height products by improving their algorithms and including more data acquisitions.

L26, about "new spin-up": What is new compared to what?

**Response:** We have deleted the word 'new'.

L53: SEIB-DFVM >> SEIB-DGVM

**Response:** We have fixed it.

L111-114: it sounds curious to me – this sentence tells the model has been already evaluated and calibrated globally, but this paper, in my understanding, tackles with the same issue. In addition, the reference of Ma et al. (2021) appears to target a specific USA domain, not global.

**Response:** We have removed it to avoid the confusion.

L161: In my impression, the subsection title "2.2 Model initialization" should be changed to "2.2 Model initialization and overview of experiments", because the second paragraph refers to transition simulation, not initialization.

**Response:** We changed the title to "2.2 Model initialization and overview of experiments".

L165: Are 1000 years enough to obtain a real equilibrium state in your model? Considering that ED model is DGVM and that the time evolution of vegetation distribution and land carbon amount depend on each other, I would have expected more than a few thousand years are required to obtain a real carbon equilibrium. If not perfectly equilibrated, it may be one of the reasons of your model to make relatively lower amount of global SOC.

**Response:** Very good point and we agree that some ESMs/DGVMs need to spin up for thousands of years in order to reach equilibrium in carbon pools. Here we add a figure of vegetation and soil carbon of equilibrium run of ED v3.0. The left two panels are total carbon in vegetation and soil pools over the last 36 years, and the right panels are respective annual changes. The figure indicates the vegetation and soil carbon are near equilibrium within the 1000 years of spin-up.

[Figure]

In terms of low global SOC, we think it is primarily due to the rapid turnover rate of SOC which could also shorten the time required to reach equilibrium for SOC.

L226: This paragraph reminds me one paper: Spafford and MacDougall (2021) GMD reviewed the processes of validation processes of land models which are coupled with ESMs. Your evaluation actually covers critical variables of carbon cycle as performed in the paper, and thus this reference may help to emphasize your rational of your variable choice.

**Response:** We agree that this paper is relevant to our choice of variables, thus we added the citation to the first paragraph of Section 2.4.

L233: I agree with the idea to aggregate the vegetation categories to compare the simulated result with satellite-derived datasets. At a same time, I wonder showing PFT distribution map with original model category would be helpful for readers and potential model users, when interpreting the simulated result. So, I would suggest putting a map with original model category in appendix.

**Response:** We agree and added the map as figure S1.

L295, about "water fluxes" ~: In the subsections of 2.4.1~2.4.4, there seems no description on the reference dataset for water flux.

**Response:** We have added a paragraph as section 2.4.4 for water fluxes.

L327: Latest assessment of global SOC stock of CMIP6 ESMs have been performed by Ito et al. 2020, ERL; global SOC of your model is still within the range of the CMIP6 ESMs, but outside the 1 S.D range. As performed by Ito et al. (2020) and Arora et al. (2020) Biogeoscience, calculation of mean residence time of SOC in global/grid scale would reveal whether such lower SOC in your model is caused by turnover rate of SOC. I believe readers can obtain a benefit from such a bit deeper analysis and the discussion.

**Response:** Very good suggestion. We calculated the residence time of SOC by dividing global total SOC by global total heterotrophic respiration, and residence time is 11.4/yr, which seems close to the lower bound of CMIP6 ESMs. We have also added the following discussion to page 14, line 477-485:

*... Second, while relative patterns for soil carbon showed close agreement at the biome level for the majority of biomes, the absolute magnitude of soil carbon was much lower than reference for several biomes and thus globally. Before over-interpreting these differences, it should be noted that there are substantial uncertainties with current empirical soil carbon maps in terms of both global totals and spatial distribution (Todd-Brown et al., 2013). Model errors in soil carbon may arise from poor representations of biophysical conditions, inaccurate parameterization, or lack of other important drivers. Soil carbon representation in ED, like that of many other DGVMs/ESMs, is highly simplified and the relatively low soil carbon is consistent with a relatively short residence time of soil carbon (about 11.4 years), which was close to the lower bound of other CMIP6 ESMs (Ito et al., 2020) ...*

L368-374: I agree that ET is an important indicator of hydrology, and it affects terrestrial carbon cycle via soil water availability, etc. In addition, transpiration is tightly connected to photosynthesis. Such importance / purpose to evaluate ET should be addressed in somewhere.

**Response:** We agree and have addressed this point through the addition of section 2.4.4 as described above.

L387, about "less agreement with the references in the tropics": the delayed timing GPP reduction in tropics (15~0 deg, Fig. 8) seems linked to the delayed timing of LAI (Fig. 14), suggesting there is some problems with the phenology scheme. This is just my speculation, and so further insights on model biases by the model developers would be helpful for readers.

**Response:** We agree that it could be related to phenology as it is a common challenge for DGVMs to capture tropical seasonality of carbon fluxes and LAI. We added more discussion on this point to page 14-15 line 489-494:

*... Fourth, the seasonality of GPP and LAI in tropics differed from reference datasets. The pattern and timing of seasonality in the tropics is scientifically challenging to understand, and has been the subject of several recent studies (Morton et al., 2014; Saleska et al., 2016; Tang et al., 2017). In ED, similar to other DGVMs/ESMs, soil water availability is assumed as the primary driver to tropical phenology. Such mechanisms lead to reduced LAI and GPP over dry seasons, which contrast to observations (Restrepo-Coupe et al., 2016).*

Fig. 16: GEDI L2B product shows discontinuous and large values in 10-15m height in all latitudinal bands, which makes as if ED model underestimate the LAI in the corresponding height. Is it possible to make discussion about whether this is derived from a systematic bias in the GEDI product, or whether it is a certain observational fact?

**Response:** Very good point. We re-examined the LAI profile comparison and updated the figure with a new version of GEDI L2B (v002). As you can see, the major change is large reduced LAI at 0-5m interval, ED and GEDI still agree that LAI decreases as height. We have revised the corresponding description in the manuscript (page 12, line 399-404).

L429-430: To my knowledge, Watanabe et al. (2011) GMD and Dunne et al. (2020) J. Of Advances in Modeling Earth systems have already incorporated ED-like models (individual-based ecosystem/carbon cycle models) with land-use change impact, into Earth system models.

**Response**: We have removed this sentence.

L459-465: In addition to the overestimation, I'm concerned about the underestimation of several variables in African savanna. When seeing the PFT distribution map (Fig. 3), the region has less vegetation, and GPP, AGB, SOC, and tree height are also underestimated. I'm wondering this model may overestimate bare ground fraction in the corresponding grids. Is this caused by LUH2 scenarios? Or fire impact? Since this paper is a kind of model description paper, further discussion by model developers on the potential reasons for the biases would be much appreciated.

**Response:** The underestimation of these variables in savanna is likely due to fire impact as it is the dominant disturbance in this area. In current transient simulations, the burned areas are prescribed by GFED. The representation of the impacts of fire on vegetation was highly

simplified, and it's possible that greater survivorship for trees could improve results. Disturbance regimes in savannas are complex, and important for future model development.

L473-474: Considering that your model reasonably reproduces GPP in boreal forest/Taiga region, NPP reproduced by the model would be well reproduced as well (if autotrophic respiration is not much biased). So, I'm wondering the potential reasons for lower SOC may be caused by less dependency on soil temperature, and/or relatively lower value of base decomposition rate. These are just my speculation, and such further discussion would be helpful for readers to obtain insights on your model behavior.

**Response:** Yes, we agree with you that dependency of soil temperature on temperature could be a cause. It could also be due to missing drivers in the current version of ED, such as topography. For example, there are several areas with SOC densities above 25kg C/m2 in high latitudes of the northern hemisphere (Fig 5a), which was not reproduced by ED. The mismatched spatial pattern is also a common issue with other DGVMs (see Todd-Brown et al., 2013 which is cited on page 14, line 480). Given these known challenges we attributed the differences to poor representation of biophysical conditions, inaccurate parameterization, or lack of other important drivers (page 14, line 481-482).

Considering this model is based on dynamic vegetation distribution, the model performance evaluated here is quite good. I encourage the authors to further improvement in future.

**Response:** Thank you so much for your encouragement.